# Numerical Investigation on Blast Response of Cold-Formed Steel Framing Protected with Functionally Graded Composite Material

**Elias Ali** [1,*] and **Fadi Althoey** [2]

1    Department of Civil and Environmental Engineering, The University of Alabama in Huntsville, Huntsville, AL 35805, USA
2    Department of Civil Engineering, College of Engineering, Najran University, Najran 1988, Saudi Arabia; fmalthoey@nu.edu.sa
*    Correspondence: elias.ali@uah.edu

**Abstract:** This paper presents a numerical simulation on the blast response of cold-formed steel (CFS) structural framing system protected with a functionally graded composite material (FGM) panel. The steel frame consists of four CFS studs, which were protected by 12.5 mm thick gypsum, aluminum composite, and FGM composite materials on both sides. The numerical simulation was performed using ABAQUS on a 1.8 m × 2.4 m, overall wall panel exposed to air blast on one side. A 1.0 kg TNT explosive charge placed at four standoff distances (R) of 1.0 m, 1.5 m, 2.0 m, and 2.5 m from the framing were investigated. The FGM board was modeled using a stepwise material variation using the power-law material function. Deformation and failure modes of the studs, as well as the protective materials, were compared to the same framing system but with different protective materials, including conventional gypsum boards and aluminum composite panels. Based on the observation from the analysis and computational simulation, the proposed protective composite material (FGM) resulted in a smaller deformation at peak overpressure at a given standoff distance (R) and local failure modes on studs. The same frame system with gypsum and aluminum panel exhibited excessive deformation as well as an early collapse of the CFS studs. This observation can lead to an alternative material solution in blast-resistant design.

**Keywords:** cold-formed steel; blast; standoff distance; TNT; functionally graded material; composite

## 1. Introduction

Over the past three decades, particularly after the Oklahoma City bombing in 1995, there has been a growing demand to incorporate blast resistance design methods and building envelope materials in government buildings and commercial facilities in the USA. Most of the works on blast-resistant materials are confined to military researches and are not easily accessible to the public. Moreover, there are only a few attempts devoted to blast-mitigation on the thin-walled structural system. An experimental study on cold-formed steel (CFS) stud wall system that uses a composite system of cement board with steel sheet as a protective sheathing was reported by Stewart et al. [1]. In their work, thorough testing of multiple stud wall systems with various spacing and connection configurations, they provided qualitative data that can be used to optimize and develop a low-cost, easily constructible wall system using the composite panels for blast resistance thin-walled system. Numerical simulations of an unprotected 20 ft ISO container exposed to a blast load of 4000 kg TNT at 120 m standoff distance using the three different approaches were conducted in a study by Børvik et al. [2]. Computational modeling of a steel stud wall system for blast-resistant design applications was developed by Bewick and Williamson [3]. In their work, they presented finite-element models that capture the peak load and deformation capacity of steel stud wall systems, accounting for the failure modes observed in past

testing. Perhaps the most recent article which focused on providing a material solution for blast resistant of the thin-walled systems was from Aviram et al. [4]. In their experimental study, they proposed a composite cement/steel sheathing material for enhanced blast resistance compared to traditional concrete or masonry wall system. In addition, this composite material can be 30% cost-effective. A report by Tao et al. [5] to AISC presented the monotonic and cyclic response of single shear cold-formed steel-to-steel and sheathing-to-steel connections. The report primarily focused on the connection detail response to dynamic loading through an experimental study and did not cover the response of the wall system. An experimental and numerical study on blast response of cold-formed steel wall performed at the Engineer Research and Development Center (ERDC) of the U.S. Army Corps of Engineers to understand the structural response of buildings subjected to blast loadings from high explosive devices such as terrorist bombs was presented by [6]. Since CFS is now in high demand for low- to mid-rise buildings, there is also a huge research interest on wall response under extreme loading environments. One of the first investigations on the response and performance of the CFS frame system protected by FGM and the studs collapse behavior under fire was conducted by two studies [7,8]. Cyclic axial response and energy dissipation of cold-formed steel members are also presented by [9]. An experimental investigation on SEB walls consisting of conventional CFS studs sheathed with a composite cement board/steel plate sheathing with enhanced detailing was investigated in [10]. Their research found out that steel stud walls constructed using conventional detailing have limited blast resistance due to premature buckling instability. A non-linear analysis of FGM sandwich plates and shell, as well as FGM large deformation analysis, is presented in [11–14]. Similar articles on FGM and resistance of steel wall systems can also be found in several studies [15–22].

In most of the articles on the response of a CFS wall system, few are devoted to providing a new material solution for building envelopes. Thus, it is the primary goal of this paper is to introduce and propose a functionally graded material (FGM) composite panel for the blast-resistant design in thin-walled framing systems. The paper investigated the blast responses of the CFS studs considering three different protective materials, several standoff distances, and FGM material function sensitivities through FE package ABAQUS. The results and observation of this study can be applied to thin-walled frames as well as to building envelopes for any building type. It can also provide a framework for blast-resistant protective material design.

## 2. Mechanical Properties of Functionally Graded Material Composite Panel

Functionally graded materials (FGMs) refer to the new class of advanced composite materials characterized by a non-homogenous material system with a gradual variation of material property within a given dimension. The concept of FGM for engineering application was first introduced in Japan in 1984 during the hypersonic spaceplane project as a thermal barrier to resist high-temperature gradients. The gradation in FGMs is achieved by either combining two or more materials using volume-fraction or chemically treating a single material to change its initial properties. The functionally graded composite material will then have a unique and different material property from the individual constituent materials while preserving their benefits. FGMs, because of their improved mechanical properties, can be useful in extreme environments such as blasts with excessive strain and high yield stresses that would cause a single-material part to fail. FGMs can also be used as structural elements, such as beams and plates; detailed analytical and numerical solutions can be found in [23,24]. The development of engineering materials leading up to FGMs is shown in Figure 1.

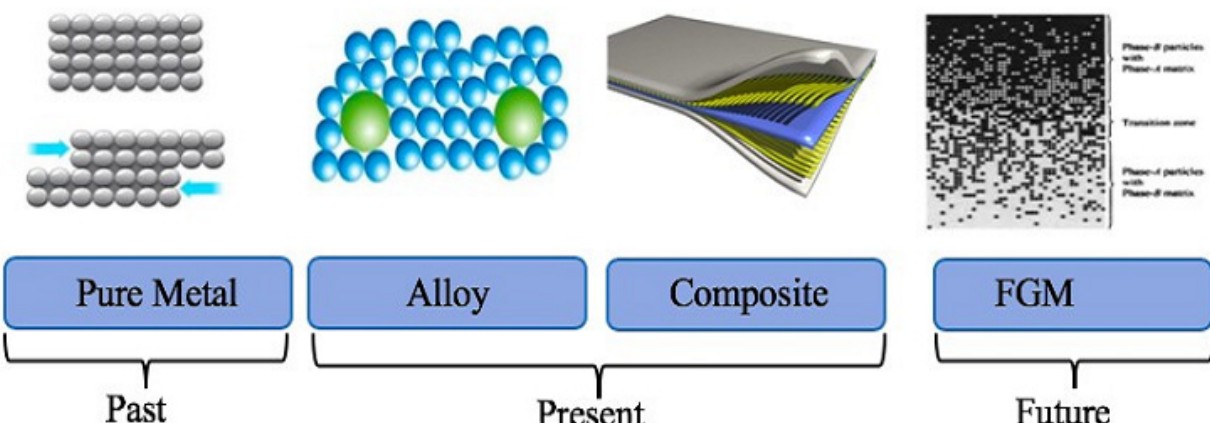

**Figure 1.** History of engineering materials leading to FGM development.

FGMs are a mixture of ceramic and metal or a combination of different metals made by gradually varying the volume fraction of the constituent materials. This volume fraction variation can be described using the power-law function, sigmoid function, or exponential function. The volume fraction variation of FGMs in the power-law function (P-FGM) within a given direction can be expressed by Equation (1):

$$f(z) = \left(\frac{z + h/2}{h}\right)^n \tag{1}$$

where z is any point within a given direction, h is the total FGM board thickness, and n is a power-law index parameter. Once the local volume fraction is defined, the functional relation of material properties at any point across the thickness can be expressed according to the general rule of mixtures. The Young's modulus variation can be calculated by using Equation (2):

$$E(z) = E_c f(z) + (1 - f(z))E_m \tag{2}$$

where $E_1$ and $E_2$ are Young's moduli of the FGM at the bottom (h/2) and top (−h/2) surfaces.

Figure 2 shows the variation of volume fraction f(z) for one of the two constituent materials forming the FGM matrix and the young's modulus variation across the thickness. This means that the second material will have a volume fraction of $1 - f(z)$ at a given location across a thickness (if the desired graduation is within a thickness direction). It can also be noticed that for every power-index value (n) considered, the FGM matrix will have 100% of one material at the top and 100% of the second material at the top.

The parent materials forming the FGM board with gradual variation are metal and ceramic with individual material properties, as shown in Table 1. Mechanical properties of the FGM board are expressed using the P-FGM function using Equations (1) and (2). Even though the Poisson ratio (υ) is also expressed using power law, its effect on the deformation across the small thickness of the panel is insignificant compared to the effect of elastic modulus. In this analysis, therefore, the υ is assumed to be constant as 0.3.

**Table 1.** Mechanical properties of FGM, gypsum board, and aluminum composite panel.

| Material | Elastic Modulus (GPa) | Yield Stress (MPa) | Density (Kg/m³) |
|---|---|---|---|
| Metal (Steel) | 210 | 420 | 7850 |
| Ceramic ($Al_2O_3$) | 390 | 260 | 2130 |
| Gypsum | 4.35 | 6.20 | 727 |
| Aluminum Composite | 69 | 110 | 2640 |

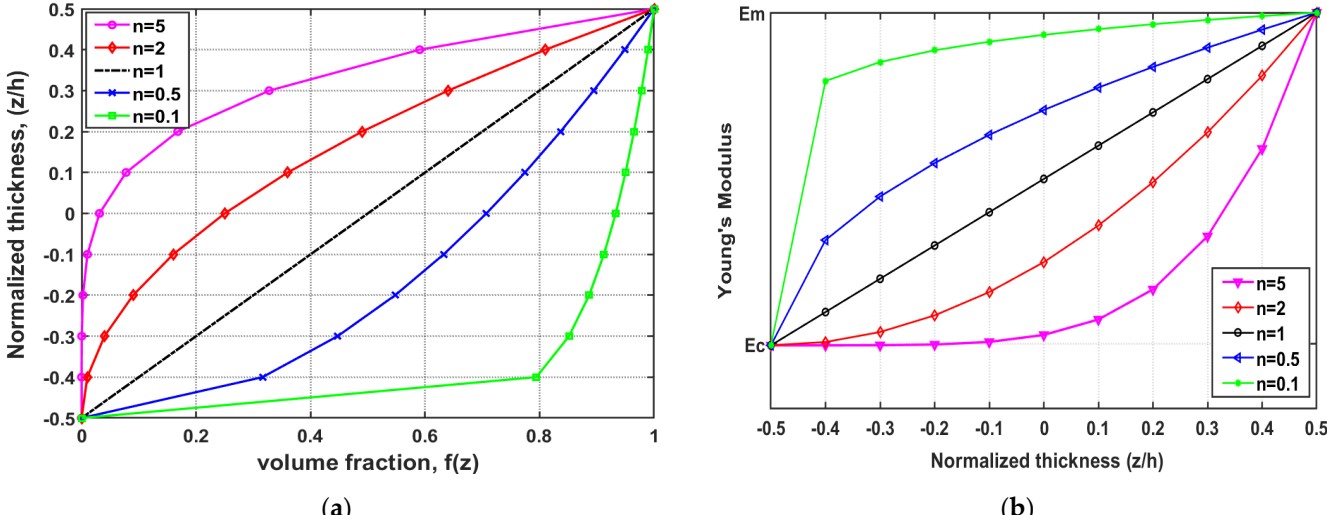

**Figure 2.** P-FGM material function (**a**) volume fraction and (**b**) Young's modulus variation.

The FGM sheathing was modeled in step-wise material graduation using eight layers while each layer across the thickness of the board was assumed to be composed of an isotropic and homogenous material based on the volume fraction defined using power law. The outer face material is 100% steel and gradually changed to 100% ceramic on the inner face of the sheathing. The choice of these two materials is based on the effect of blast loading, which results in both high stress and temperature on the board. Steel has higher tensile stress but lowers thermal resistivity, while ceramic has higher thermal resistivity with lower stress resistance.

### 3. Blast Load FE Modeling of the CFS Wall System

*3.1. CFS Wall Geometry*

The blast response of the thin-walled FGM system composed of CFS studs was investigated for three different wall cases of 12.5 mm thick FGM composite, aluminum composite panel, and gypsum plasterboard. The CFS framing was composed of four lipped channel studs 400S200-54 (101.6-mm web, 50.8-mm flange, and 1.41-mm thick with a 12.7-mm lip) spaced at 600 mm. This wall configuration was selected from the current typical wall configuration practice, as shown in Figure 3.

The overall wall panel modeled as shown in Figure 4 was 1.8 m × 2.4 m exposed to air blast on one side. Both the CFS studs and sheathing panels were modeled using an S4R-element type with a mesh size of 1.0 cm and 5.0 cm for the studs and the panels respectively. Due to the computational time and storage capacity needed to simulate the entire wall, the overall wall height is reduced to 1.2 m instead of 2.4 m, which reduces mesh and element numbers. Both ends of the studs are assumed to have a pinned-pinned boundary condition with additional restraining support from the bolt connection at equal spacing (300 mm). The interaction between the sheathing materials (gypsum, aluminum, and FGM) and the connection was modeled using the Beam MPC constraint, which resulted in restaining of all the six degrees of freedom at the connection. Figure 3 represents the mesh and geometry of the model used in ABAQUS in this study.

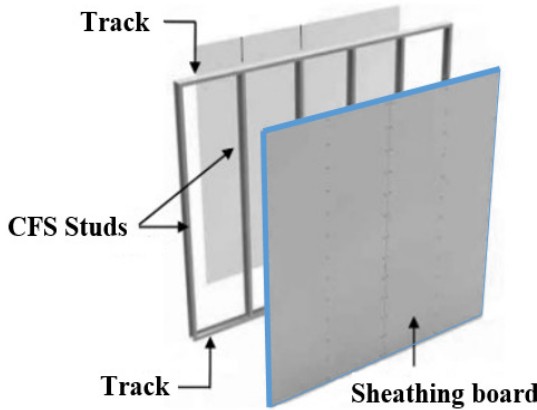

**Figure 3.** Typical thin-walled structural wall configuration.

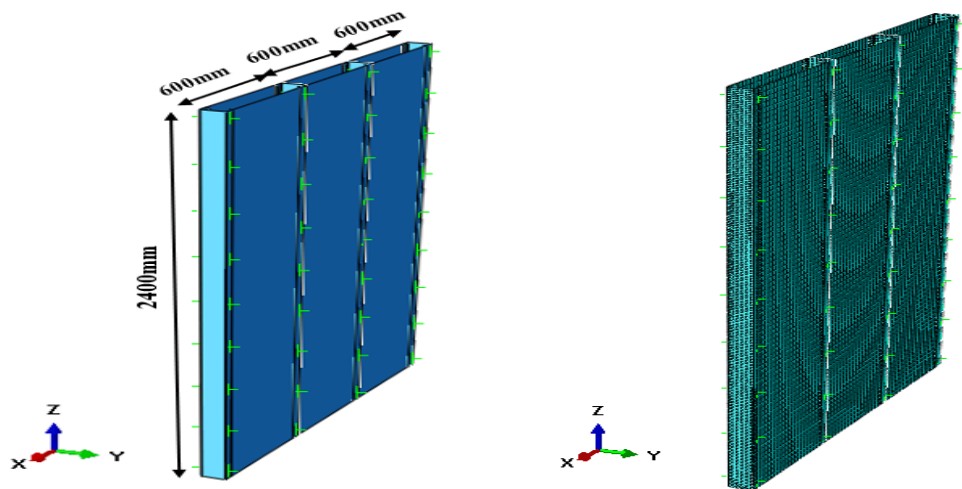

**Figure 4.** CFS Stud wall panel configuration and FE mesh detail.

### 3.2. FGM Board Modeling

The proper identification of material properties for each sheathing board is crucial for the reliability of the analysis. The FGM material properties (elastic modulus, yield stress, and mass density), which are spatially varying along the thickness of the board were carefully modeled in the FE simulation. These property changes along the eight layers (ply) are shown in Table 2. Since the current commercial software's like ABAQUS concentrates on approximate modeling using piecewise homogeneous material assignment, the FGM sheathing in this study was modeled using step-wise material graduation considering eight layers across the sheathing thickness. Each layer was assumed to be isotropic and homogenous material based on the volume fraction defined using the power-law function stated in Equations (1) and (2). The outer face material is 100% steel and gradually changed to 100% ceramic on the inner face of the sheathing.

In the FE model, the studs and the sheathing boards (gypsum, aluminum, and FGM board) were connected using fasteners at eight equally spaced screw locations, along the flanges of studs spaced at 30 cm, as shown in Figure 3. Connectors were modeled as rigid beams, by tying nodes at the center of CFS flanges and adjacent nodes on the boards, within a radius of 2.0 cm.

**Table 2.** Mechanical properties of FGM used in FE Modeling.

| Ply No. | Elastic Modulus (pa) | | | Mass Density (kg/m$^3$) | | |
|---|---|---|---|---|---|---|
| | n = 5 | n = 1 | n = 0.1 | n = 5 | n = 1 | n = 0.1 |
| 1 (metal) | $2.03395 \times 10^{11}$ | $2.03395 \times 10^{11}$ | $2.03395 \times 10^{11}$ | 7850 | 7850 | 7850 |
| 2 | $2.89418 \times 10^{11}$ | $2.25471 \times 10^{11}$ | $2.05738 \times 10^{11}$ | 5063.839 | 7135 | 7774.128 |
| 3 | $3.38091 \times 10^{11}$ | $2.47547 \times 10^{11}$ | $2.08404 \times 10^{11}$ | 3487.383 | 6420 | 7687.79 |
| 4 | $3.63158 \times 10^{11}$ | $2.69622 \times 10^{11}$ | $2.11504 \times 10^{11}$ | 2675.502 | 5705 | 7587.378 |
| 5 | $3.7869 \times 10^{11}$ | $3.13773 \times 10^{11}$ | $2.19895 \times 10^{11}$ | 2172.418 | 4275 | 7315.602 |
| 6 | $3.79828 \times 10^{11}$ | $3.35849 \times 10^{11}$ | $2.26257 \times 10^{11}$ | 2135.586 | 3560 | 7109.549 |
| 7 | $3.79995 \times 10^{11}$ | $3.57924 \times 10^{11}$ | $2.36552 \times 10^{11}$ | 2130.175 | 2845 | 6776.084 |
| 8 (ceramic) | $3.8 \times 10^{11}$ | $3.8 \times 10^{11}$ | $3.8 \times 10^{11}$ | 2130 | 2130 | 2130 |

*3.3. Blast Loading*

Blast (explosion) can be defined as a rapid, large scale, and sudden release of energy from a given source categorized based on the physical state as solids, liquids, or gases. The blast wave-time history shown in Figure 5a indicates that when an explosive detonates, a layer of compressed air (blast wave) is formed, resulting in an instantaneous pressure increase ($P_{so}$) above the ambient atmospheric pressure ($P_o$) at an arrival time ($t_A$). This overpressure then decays to ambient atmospheric pressure at time td; it further decays to under pressure ($-P_{so}$), creating a partial vacuum, before it finally returns to ambient atmospheric pressure. The area under the curve for the positive duration is called a specific impulse. In this study, Conwep blast modeling was adopted using ABAQUS to simulate this overpressure considering an equivalent explosive TNT weight of 1.0 kg for all simulations. The standoff distance (R) of 1.0 m, 1.5 m, 2.0 m, and 2.5 m from the walls were considered to simulate a dynamic response of the wall systems, as shown in Figure 5b.

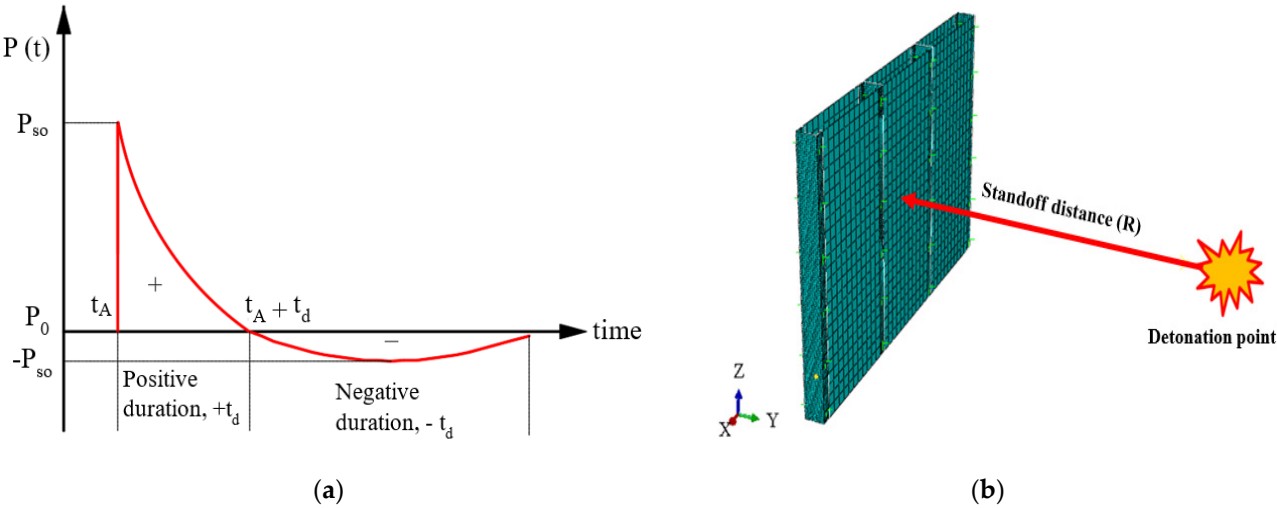

| (a) | (b) |

**Figure 5.** Blast loading (**a**) Blast pressure profile and (**b**) FE wall model under blast loading.

**4. Results and Discussion**

FE results from the dynamic explicit analysis are presented in Figures 6–9 for the CFS wall system protected by the FGM composite, aluminum composite panel, and conventional gypsum boards. The wall system protected with gypsum sheathing exhibits higher deformation and early CFS stud failure at all standoff distances considered for TNT charge (1.0 kg) compared to those protected with FGM sheathing and aluminum composite panel.

Figure 7 shows the stress and buckling modes from a blast at a standoff distance of 2.5 m for the CFS frame protected with a gypsum board. The CFS center studs in gypsum sheathing exhibited mainly local buckling with yielding at the connection while CFS studs

at the two ends exhibited distortional buckling in all standoff distances considered in the analysis.

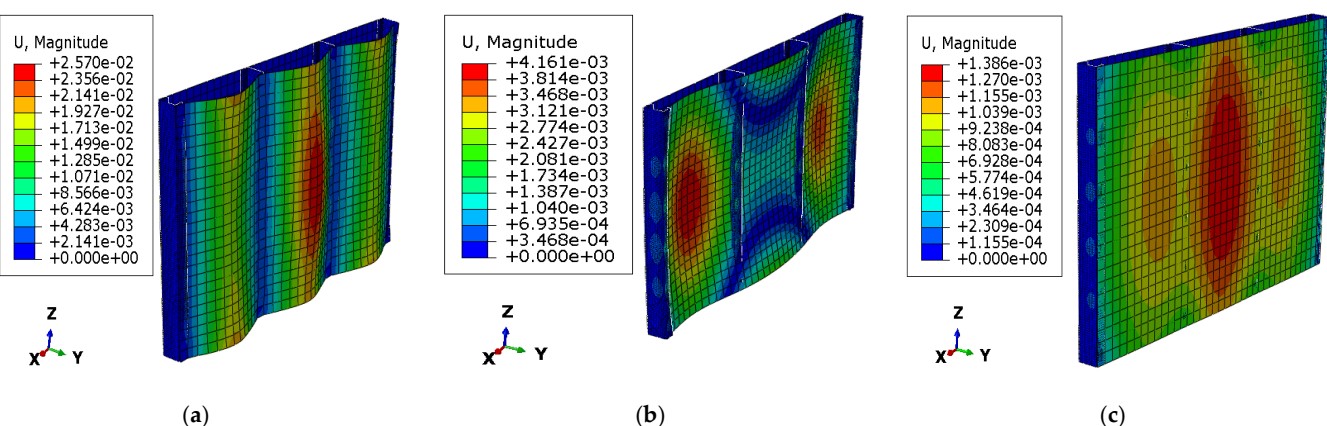

**Figure 6.** Deflection on the CFS framing with (**a**) gypsum board, (**b**) aluminum composite, and (**c**) FGM composite for a blast at R = 2.5 m.

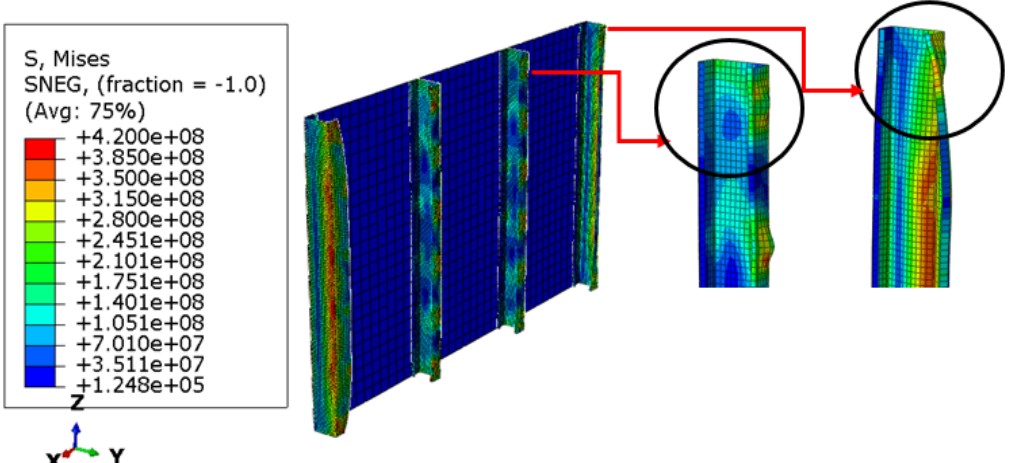

**Figure 7.** Stress and buckling modes of CFS studs with gypsum board at R = 2.5 m.

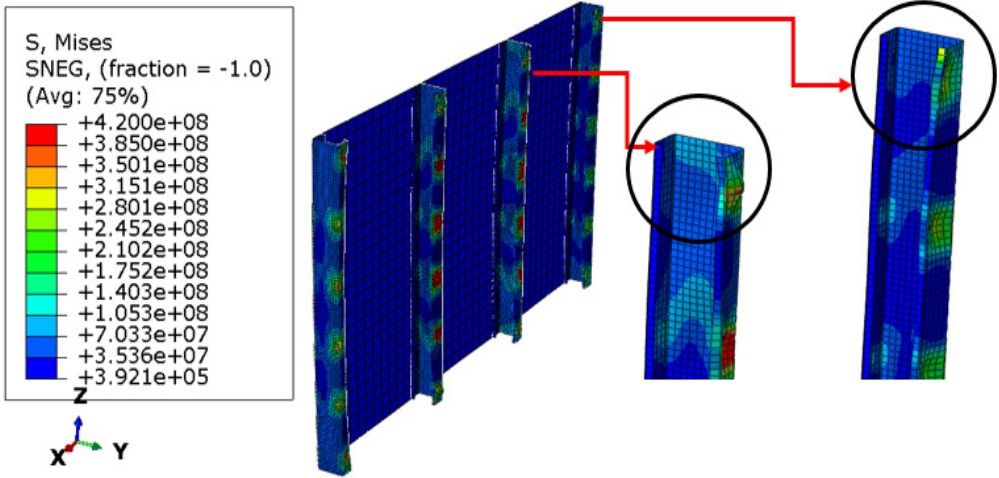

**Figure 8.** Stress and buckling modes of CFS studs with an aluminum composite panel at R = 2.5 m.

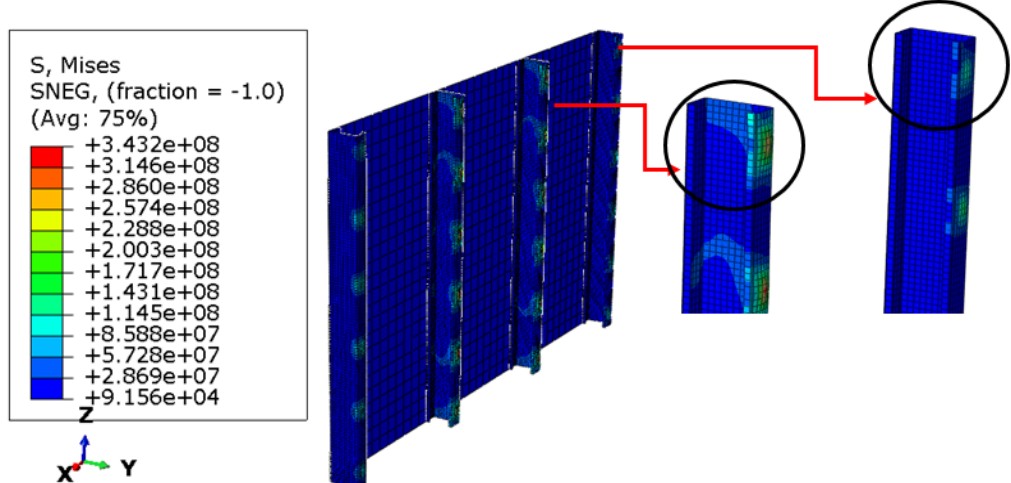

**Figure 9.** Stress and buckling modes of CFS-studs with FGM composite panel n( = 0.1) at R = 2.5 m.

Figure 8 shows the stress and buckling modes from a blast at a standoff distance of 2.5 m for the CFS frame protected with an aluminum composite panel. The CFS studs exhibit a local buckling and collapse (yielding) starting at the connection between the studs and the panel. One can also notice that the restraining effect from the bolt is considered until the full analysis, which prohibits both distortional and global buckling modes at failure. Another reason can also be attributed to the small TNT charge used.

Figure 9 shows the stress and buckling modes from a blast at a standoff distance of 2.5 m for the CFS frame protected with FGM composite material. The CFS wall showed the smallest deformation from all blast standoff distances compared with gypsum and aluminum panel framings, and the CFS studs remain below the yield stress (420 MPa) with local buckling at connections on all four studs for a blast at a stand-off distance of 2.5 m. The maximum stress, in this case, was 343.2 MPa at the connection. Furthermore, the connection between the studs and the FGM sheathing did not reach the yielding stress, which means that the frame resists the applied blast load without failure in part or whole.

### 4.1. FGM Material Variation Index Sensitivity

Material sensitivity was also performed in this study to investigate the effect of material variation in the P-FGM material function using three power-law index values. Figure 10 shows the displacement of the FGM sheathing at standoff distances of 1.0 m and 2.5 m. It can be observed that the FGM panel with a power-law index of 0.1 resulted in the smallest deformation compared to one with n = 1.0 and n = 5.0. The reason for such response is that steel (metal) is the predominant material that forms the FGM matrix along with the panel thickness, which resulted in higher stiffness. On the other hand, the FGM matrix with n = 5.0 is predominantly composed of ceramic with a small volume fraction of metal along with the thickness, which resulted in the smallest stiffness during blast pressure.

### 4.2. Standoff Distance Sensitivity

The blast responses of the CFS framing sheathed with gypsum board, aluminum panel, and FGM composite at four stand-off distances are shown in Figure 11. This was investigated to develop a safe distance to the framing system before the collapse. For that, a total of 12 dynamic analyses were performed. It can be observed that the CFS frame with gypsum board resulted in excessive deformation and collapse in all standoff distances considered. The CFS frame with the FGM composite panel exhibited the smallest deformation in all cases while remaining below the yielding stress during the blast with a TNT charge of 1.0 kg. The CFS frame with aluminum composite panel showed the second least deformation, however, the CFS studs reached yielding stress and resulted in a collapse in all cases considered.

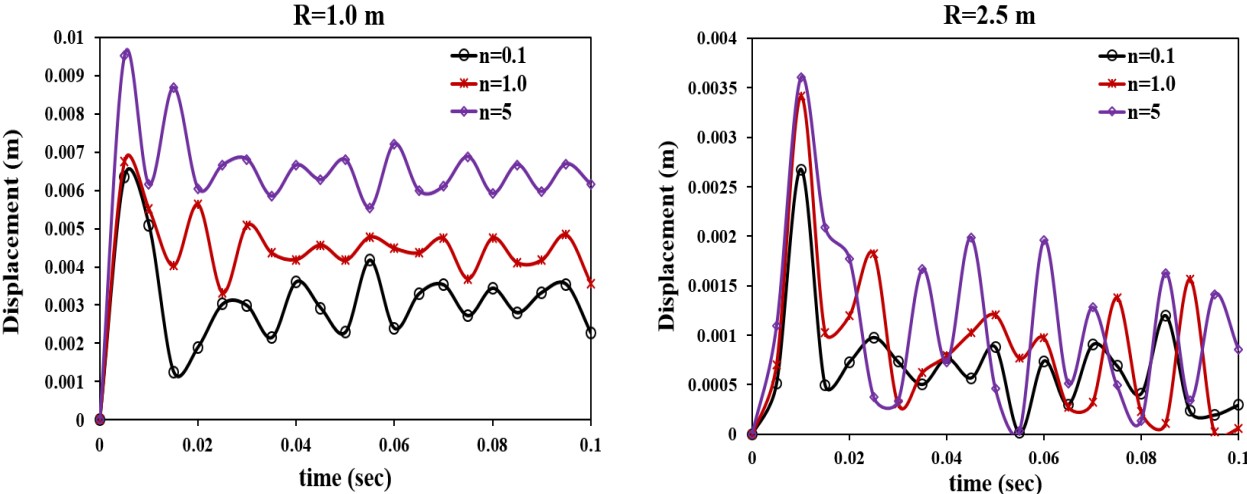

**Figure 10.** Effect of the power-law index on the deformation of FGM sheathing.

The comparison presented here demonstrated that the FGM sheathed thin-walled system can withstand a 1.0 kg TNT charge before failure at a 2.5 m detonation point and further. It is believed that this original idea on the use of FGMs for blast-resistant wall systems could provide new insight and be a starting ground for future research and possible future application in the blast-resistant design of thin-walled structures.

**Figure 11.** Standoff distance sensitivity on the three sheathing materials.

## 5. Conclusions

The paper provide a state-of-the-art numerical investigation of the new composite material in the blast-resistant deign of thin-walled framing consiting of CFS studs. The dynamic explicit analysis on the blast response of the cold-formed steel (CFS) wall system protected by conventional gypsum board, aluminum composite panel, and FGM composite was investigated against equivalent explosive TNT weight of 1.0 Kg. The blast standoff distances (R) of 1.0 m, 1.5 m, 2.0 m, and 2.5 m from the wall were considered to simulate the dynamic response of the wall systems. It was observed that the CFS wall system with gypsum sheathing exhibited excessive deformation and early CFS stud failure, even for a very small TNT charge and at all standoff distances considered herein compared to those with FGM sheathing and aluminum composite.

The CFS studs in the gypsum sheathed wall exhibit a local buckling and collapse starting at the connection between the studs and sheathing materials. The CFS studs in aluminum composite exhibited mainly local buckling and reached yielding stress (420 MPa) at all standoff distances, while studs in the FGM wall showed the smallest deformation and resulted in only elastic stress (343.2 MPa) for the blast at a standoff distance of 2.5 m. Furthermore, the connection between the studs and the FGM sheathing did not reach yielding stress.

The results presented in this paper provide a starting ground for future research on the use of mutli-functional materias for extreme loading environments. However, to capture the whole behavior under blast loading, there should be a further FE analysis and experimental investigation to include a parametric study on the size of detonation (TNT charge), detonation point, and influence of connection on the response of the wall system.

**Author Contributions:** Conceptualization, E.A.; methodology, E.A.; software, E.A.; validation, E.A. and F.A.; writing—original draft preparation, E.A.; writing—review and editing, F.A. All authors have read and agreed to the published version of the manuscript.

**Funding:** This research received no external funding.

**Institutional Review Board Statement:** Not applicable.

**Informed Consent Statement:** Not applicable.

**Conflicts of Interest:** The authors declare no conflict of interest.

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
