# Peer review of "Numerical Investigation on Blast Response of Cold-Formed Steel Framing Protected with Functionally Graded Composite Material"

_buildings, doi:10.3390/buildings12020118_

Round 1

Reviewer 1 Report

Dear Authors,

In the article there is presented a very interesting problem of numerical simulation on the blast response of cold-formed steel (CFS) structural framing system protected with a functionally graded composite material (FGM) panel. To make the content of the article more clear to a reader authors should take into consideration the following points:

1. No description of material models used in ABAQUS. There is no information on the verification of the finite element mesh.

2. There is no exact description of the FGM material used in the program. What properties did it have? Which properties were entered into the program?

3. Please indicate exactly what is the power-law index "n".

4. Why the article presents only the results of stresses and displacements for the distance of 2.5 m. At a closer distance, the effects of failure will be more pronounced. Please explain and complete.

5. Final conclusions require expansion with additional energies, etc. features describing the explosion. A lot of such information can be obtained from the ABARUS program.

6. Literature should be expanded, for example, to include scientists from Central and Eastern Europe. For example, sandwich panel testing.

Author Response

Thank you for the critical review. Please see the attachment to view our responses.

Reviewer 2 Report

This paper has been devoted to study the dynamic behavior of cold-formed steel frame protected by FGM under blast loading. Only numerical study has been done in this research. A comparison study has been implemented between two methods of protection including composite FGM and conventional gypsum along with aluminum composite panels. The research is interesting but the novelty is not sufficient. The authors should clarify some issues discussed below:

1- The main contribution of the research should be clarified. What is the main novelty of this work? What is the main application of the results obtained by this study?

2- The abstract of the paper should be re-organized. Important topics including the highlights of the study should be presented in this part. In addition, some effective topics about geometric and material properties of the model on the obtained results should be presented.

3- The details of composite FGM should be discussed in the abstract. What do you mean "functionally graded composite material"? Do you mean "Functionally Graded Material (FGM)"? This issue should be also considered in the title of the paper.

4- At the first paragraph of the introduction, the authors should clarify justifications to use this method of protection. According to the cost of FGM, is it logical to use this type of material for protection? In which type of buildings, using this type of material is affordable?

5- The literature review of the article should be improved by adding some new numerical studies on the behavior of structures composed of FGMs. The authors can found some useful information in the following references: Fire Safety Journal 125 (2021): 103425-World Journal of Engineering 16 (5), 636-647, 2019-Applied Sciences 11 (24), 11747-https://doi.org/10.1080/15376494.2020.1780524-Steel and Composite Structures 39 (5), 599-614, 2021.

5- In the last paragraph of the introduction, the steps of the research should be completely discussed. This part should be re-written.

6- Why did the authors use power function for distribution pattern of the materials?

7- Only elastic modulus of the material has been functionally distributed. Poisson's ratio and density should be also functionally distributed. 

8- Details of modeling FGM in ABAQUS should be presented by the authors.

9- In addition, the details of modeling including mesh type, mesh size and analysis type in the ABAQUS should be discussed completely. 

10- what is the difference between Figure 2(a) and Figure 2(b)? It seems that they are the same.

11- In Eq. (2), it is better to use Em and Ec instead of E1 and E2. In addition, please clarify when n=0 , we have cermaic or metal.

12- There are some other references in which the behavior of laminate shell structures composed of FGM is investigated. It is better to study them in the literature, too. Such as: Aerospace Science and Technology 87, 178-189, 2019

13- The conclusion part has not been well-organized. It is better to re-write this section. A summary of novelty and highlights of the work should be discussed in this part, first. Then, the significant results which have been only obtained by this study should be presented. 

Author Response

(The authors gave the same response as above.)

Round 2

Reviewer 2 Report

All comments have been considered by the authors correctly.